# A 2-Year Audit on Antibiotic Resistance Patterns from a Urology Department in Greece

**DOI:** 10.3390/jcm12093180

**Published:** 2023-04-28

**Authors:** Ioannis Manolitsis, Georgios Feretzakis, Stamatios Katsimperis, Panagiotis Angelopoulos, Evangelos Loupelis, Nikoleta Skarmoutsou, Lazaros Tzelves, Andreas Skolarikos

**Affiliations:** 1Second Department of Urology, Sismanogleio General Hospital, 15126 Marousi, Greece; 2School of Science and Technology, Hellenic Open University, 26335 Patras, Greece; 3Department of Quality Control, Research and Continuing Education, Sismanogleio General Hospital, 15126 Marousi, Greece; 4IT Department, Sismanogleio General Hospital, 15126 Marousi, Greece; 5Microbiology Laboratory, Sismanogleio General Hospital, 15126 Marousi, Greece

**Keywords:** urinary tract infections, antimicrobial resistance, antibiotics, multidrug resistance, urological infections

## Abstract

Purpose: The high incidence of urinary tract infections (UTIs), often in nosocomial environments, is a major cause of antimicrobial resistance (AMR). The dissemination of antibiotic-resistant infections results in very high health and economic burdens for patients and healthcare systems, respectively. This study aims to determine and present the antibiotic resistance profiles of the most common pathogens in a urology department in Greece. Methods: During the period 2019–2020, we included 12,215 clinical samples of blood and urine specimens that tested positive for the following pathogens: *Escherichia coli*, *Enterococcus faecium*, *Enterococcus faecalis*, *Proteus mirabilis*, *Klebsiella pneumoniae*, or *Pseudomonas aeruginosa*, as these are the most commonly encountered microbes in a urology department. Results: The analysis revealed a 22.30% mean resistance rate of *E. coli* strains with a 76.42% resistance to ampicillin and a 54.76% resistance rate to ciprofloxacin in the two-year period. It also showed an approximately 19% resistance rate of *P. mirabilis* strains and a mean resistance rate of 46.205% of *K. pneumoniae* strains, with a decreasing trend during the four semesters (*p*-value < 0.001), which presented an 80% resistance rate to ampicillin/sulbactam and 73.33% to ciprofloxacin. The resistance to carbapenems was reported to be 39.82%. The analysis revealed a 24.17% mean resistance rate of *P. aeruginosa* with a declining rate over the two-year period (*p*-value < 0.001). The *P. aeruginosa* strains were 38% resistant to fluoroquinolones and presented varying resistance against carbapenems (31.58% against doripenem and 19.79% against meropenem). Regarding the *Enteroccocal* strains, a 46.91% mean resistance was noted for *E. faecium* with 100% resistance to ampicillin, and a 24.247% mean resistance rate for *E. faecalis* strains that were 41% resistant to ciprofloxacin. Both types showed 100% sensitivity to linezolid. Conclusions: The dissemination of antibiotic-resistant pathogens poses the need to implement surveillance programs and, consequently, to develop strategies to prevent the emergence of such pathogens in order to optimize patient outcomes.

## 1. Introduction

Urinary tract infections (UTIs) are one of the most common infectious diseases, accounting for 150–250 million cases globally every year [1]. They occur frequently, in the context of a nosocomial environment, leading to an increasing antimicrobial resistance (AMR) [2]. AMR can emerge through a number of methods. As with fluoroquinolone resistance, which is primarily brought on by changes in DNA-topoisomerase, mutations may occur in the target of the antimicrobial agent. As an alternative, the medication may be inactivated by an enzyme, as in the case of beta-lactamases, or by an aminoglycoside through phosphorylation, acetylation, or adenylation. Two other methods include evading the target, as in the case of vancomycin, and preventing the medicine from reaching its intended targets, as in the example of tetracycline [3,4].

AMR has become a major concern in medicine in the twenty-first century. The dissemination of antibiotic-resistant infections constitutes a worldwide concern with an increasing health and economic burden [5]. The World Health Organization (WHO) is currently considering this issue to be “one of the biggest threats to global health, food security, and development”. The major project undertaken by the WHO to track worldwide AMR and antibiotic usage since 2015 is the worldwide Antimicrobial Resistance and Use Surveillance System (GLASS). In terms of uropathogenic microorganisms, GLASS gathers information on the AMR rates of *Escherichia coli* and *Klebsiella pneumoniae*. The report from the year 2021 shows a median AMR against cotrimoxazole of 54.4% (IQR 36.5–69.4) for E. coli and 43.1% (IQR 31.8–57.5) for *K. pneumoniae* in 12 countries reporting globally [6].

According to data from the European Antimicrobial Resistance Surveillance Network (EARS-Net), Greece has one of the highest burdens of infections caused by antibiotic-resistant bacteria [7]. The majority of this burden was caused by infections with carbapenem-resistant or colistin-resistant bacteria in 2015 [2,7,8]. One of the reasons why Greece ranks high in antibiotic-resistant bacteria is self-medication, as antibiotics are acquired very often from pharmacies without any prescription. In addition, the over-prescription of antibiotics by Greek physicians is a major issue that leads to antibiotic over-consumption and, consequently, to an increasing antimicrobial resistance [9,10].

Antimicrobial stewardship programs aim to optimize the appropriate use of available antimicrobial agents in order to improve therapeutic outcomes for multidrug-resistant (MDR) infections, thereby lowering antimicrobial resistance rates and hospital costs [11]. To implement any preventive measures, it is critical to understand the antibiotic resistance profiles in each hospital ward. The current study aims to determine and present the antibiotic resistance levels of *E. coli*, *P. mirabilis*, *E. faecium*, *E. faecalis*, *P. aeruginosa*, and *K. pneumoniae* isolates over a two-year period, 2019–2020, in one of Greece’s largest urology departments.

## 2. Materials and Methods

This was a cross-sectional study conducted in Sismanogleio General Hospital, a public tertiary hospital in Athens, Greece, based on data from the Microbiology Department.

### 2.1. Study Design

The inclusion criteria of this study included clinical isolates from non-identical, hospitalized patients that were not duplicated. On the other hand, we excluded all isolates that originated from a single patient with similar susceptibility patterns. We included *E. coli*, *E. faecium*, *E. faecalis*, *P. mirabilis*, *K. pneumoniae*, and *P. aeruginosa* as they constitute the most common pathogens encountered in a urology department [12]. Resistance to the following antibiotics was measured in *P. aeruginosa*: aminoglycosides (amikacin, gentamycin, and tobramycin), fluoroquinolones (ciprofloxacin and levofloxacin), cephalosporins (cefepime and ceftazidime), monobactams (aztreonam), carbapenems (doripenem, meropenem, and imipenem), piperacillin/tazobactam (anti-pseudomonal penicillin), and colistin. The following antibiotics were used to assess *E. coli* resistance: aminoglycosides (amikacin and gentamycin), aminopenicillins (amoxicillin/clavulanic acid and ampicillin), cephalosporins (cefuroxime and ceftazidime), carbapenems (meropenem), trimethoprim/sulfamethoxazole, and piperacillin/tazobactam. The antibiotics used to test *K. pneumoniae* resistance were aminoglycosides (amikacin, gentamycin, and tobramycin), aminopenicillins (ampicillin/sulbactam), cephalosporins (cefotaxime, ceftazidime, and cefuroxime), fluoroquinolones (levofloxacin and moxifloxacin), tetracycline, and colistin. The resistance for *P. mirabilis* was measured based on the following antibiotics: aminoglycosides (amikacin and gentamycin), aminopenicillins (amoxicillin/clavulanic acid and ampicillin), cephalosporins (cefotaxime, cefoxitin, ceftazidime, and cefuroxime), carbapenems (imipenem and meropenem), fluoroquinolones (levofloxacin), piperacillin/tazobactam, and trimethoprim/sulfamethoxazole. As for the *Enterococci* species, the resistance for *E. faecium* and faecalis was measured based on ampicillin, fluoroquinolones (ciprofloxacin and levofloxacin), aminoglycosides (gentamycin), glycopeptide antibiotics (vancomycin and teicoplanin), lipopeptide antibiotics (daptomycin), oxazolidinones (linezolid), streptogramins (pristinamycin), and tetracycline. Our research focused on the antibiotics mentioned above since there were an adequate number of samples for these in order to extract reliable conclusions and there was also susceptibility testing performed for them in all 4 semesters.

### 2.2. Source of Isolates

During a 2-year period (January 2019 to December 2020), a total of 12,215 clinical samples from patients hospitalized in the urology department were included in this study and processed by the Microbiology Laboratory according to established protocols [13,14,15]. The samples included blood and urine specimens. Positive urine cultures were classified as having considerable microbial growth, as indicated by standard microbiological criteria [16]. Blood cultures were collected from all febrile patients that were hospitalized in our department. Blood cultures were incubated in the BacT/Alert system (bioMerieux, Marcy-l’Étoile, France). Isolation and identification of pathogens was carried out according to classical microbiological procedures, such as conventional cultivation-based enrichment and plating methods with rapid plating solution [17].

### 2.3. Antimicrobial Susceptibility Testing

Antimicrobial susceptibility testing was carried out using the MicroScan system (Siemens, Munich, Germany) in accordance with CLSI guidelines [18,19], and the results were confirmed using a gradient MIC (minimum inhibitory concentration) method in accordance with the manufacturer’s guidelines (e.g., the E-test bioMerieux, Solna, Sweden). Colistin MICs were retested using microtiter plates (SensiTestColistin, Liofilchem, Roseto degli Abruzzi, Italy). CLSI criteria were used to determine antibiotic sensitivity and resistance breakpoints [18,19]. As quality control strains for susceptibility tests, *Escherichia coli* ATCC 25922 and *Pseudomonas aeruginosa* ATCC 27853 were used. The CLSI-recommended double-disk synergy test (DDST) was used to detect the phenotype of the production of extended-spectrum beta-lactamases (ESBL).

Metallo-beta-lactamases (MBL) and carbapenemases (KPC) were detected phenotypically by (a) the modified Hodge test [19]; (b) the combined disk test, with a meropenem (MER) disk alone, a MER disk plus phenyl boronic acid (PBA), a MER disk plus EDTA, and a MER disk plus PBA and EDTA [20]; and (c) the NG CARBA 5 immunochromato-graphic assay, targeting KPC-, NDM-, VIM-, and IMP-type and OXA-48-like car-bapenemases, following the manufacturer’s guidelines [21]. *P. aeruginosa* strains were phenotypically tested for MBL using either an imipenem (IPM) disk and IPM plus EDTA combined disk test [22] or an IPM-EDTA double-disk synergy test (DDST) [23].

PCR testing was performed for the detection of the most clinically significant carbapenemase genes (blaNDM, blaVIM, blaKPC, and blaOXA-48), as they encode enzymes that may generate carbapenem and other beta-lactam resistance. The testing was carried out on all strains that phenotypically produced more than one or no carbapenemases, as well as for the OXA producers and all those tested with NG CARBA 5. They were also tested for the presence of the colistin-resistance plasmid-mediated mcr-1 gene [21,24].

### 2.4. Statistical Analysis

All analyses were performed using SPSS Statistics version 24.0 (IBM, Armonk, NY, USA) [25]. The data were analyzed over a period of two years (2019–2020) and each semester was analyzed in a separate manner. The resistance rates of *P. aeruginosa*, *E. coli*, *P. mirabilis*, *E. faecium*, *E. faecalis*, and *K. pneumoniae* clinical isolates against individual antimicrobial agents were analyzed. The measure of the mean resistance of any microbe is defined as the percentage of samples of a specific microbe that are resistant to any of the antibiotics that were tested in the laboratory. Differences between means regarding the resistance of those six bacteria mentioned above, in the urology department, were determined by univariate analysis of variance and Tukey’s multiple range tests. A significance level of *p* < 0.05 was used.

## 3. Results

We analyzed the resistance rates in 5430 tests of *E. coli*, 1537 tests of *E. faecalis*, 1315 tests of *P. mirabilis*, 564 tests of *E. faecium*, 2302 tests of *K. pneumoniae*, and 1067 tests of *P. aeruginosa*, against routinely tested antimicrobial agents. The mean age of our patients was 67.7 years of age, and regarding their sex, 59.2% of them were males and the remaining 40.8% were female patients.

The mean resistance rate of *E. coli* during the years 2019–2020 was 22.30%, ranging from 19.34 to 26.76% (*p*-value 0.977) (Figure 1), with the number of isolates being decreased in 2020 compared to 2019 (1739 from 3.691). The *E. coli* strains were 76.42% resistant to ampicillin, while there was 96.67% sensitivity to amikacin, and 94.09% sensitivity to piperacillin/tazobactam. The resistance rate to ciprofloxacin was reported at 54.76% during the same period (Figure 2).

Regarding *P. mirabilis*, the 2-year mean resistance rate was 18.94% with no statistical significance between the four semesters (*p*-value 0.982) (Figure 1), and approximately twice the number of isolates between the two years (807 in 2019 and 508 in 2020). As for the resistance profile of *P. mirabilis*, a 52.99% resistance to ampicillin was noted, while the sensitivity to other beta-lactams was 92.8% for cefoxitin and 83.05% for cefuroxime. The resistance to fluoroquinolones was reported at 24.53%, while there was 100% sensitivity to carbapenems (Figure 3).

For the same two-year period, the resistance rate of *K. pneumoniae* was 46.205% on average, decreasing from 55.86% in the first semester of 2019 to 30.93% in the second semester of 2020 (*p*-value < 0.001) (Figure 1), with the number of isolates also decreasing during the same period. The strains of *K. pneumoniae* were reported to be mostly resistant to ampicillin/sulbactam at 80% and ciprofloxacin at 73.33%. The resistance to other beta-lactams was 60% for cefuroxime, 61.59% for amoxicillin/clavulanic acid, 45% for piperacillin/tazobactam, and 39.82% for carbapenems. These specific strains were 33.57% resistant to amikacin, while an 86.71% sensitivity rate to colistin was noted (Figure 4).

The mean resistance rate of *P. aeruginosa* during the same period was 24.17%, decreasing from 35.98% in the first semester of 2019 to 17.70% in the second semester of the year 2020 (*p*-value < 0.001) (Figure 1), while the number of isolates originating from the urology department also decreased in the same period (682 in 2019 and 385 in 2020). A high resistance rate of 38% was reported against fluoroquinolones, while the *P. aeruginosa* strains were 30% resistant to amikacin and presented varying resistance against carbapenems (31.58% to doripenem and 19.79% to meropenem). In addition, a 77% sensitivity rate to piperacillin/tazobactam was reported in our study (Figure 5).

Regarding the *E. faecium* strains, the resistance rate was set at 46.91%, with no significant variations between semesters (Figure 1). The number of *E. faecium* isolates also presented small changes in the same period. This type of *Enteroccoci* presented 100% resistance rate to both ampicillin and fluoroquinolones, while there was almost 80% sensitivity to vancomycin and 100% sensitivity to linezolid (Figure 6). Concerning the *E. faecalis* isolates, the resistance rate was reported to be 24.247%, ranging from 23.99 to 25.51% during the period 2019–2020 (*p*-value 1) (Figure 1). In contrast to *E. faecium* isolates, these specific enterococci presented 100% sensitivity to ampicillin and 41% resistance to fluoroquinolones, while there was also 100% and approximately 99% sensitivity to linezolid and vancomycin, respectively (Figure 7).

## 4. Discussion

Antibiotic consumption is a major contributor to the spread of AMR, with Cyprus, France, Italy, and Greece being the top four antibiotic prescribers in Europe [5]. According to a study conducted by the European Centre for Disease Prevention and Control (ECDC), more than 50% of *E. coli* and over 30% of *K. pneumoniae* isolates were resistant to at least one antimicrobial group under monitoring, and combination resistance to multiple antimicrobial groups was common. Consequently, approximately 33,000 people die each year as a result of an infection caused by bacteria resistant to antibiotics [7]. Furthermore, healthcare-associated infections account for the majority of the burden of these multidrug-resistant infections, while last-line treatments such as carbapenems and colistin are becoming less effective, limiting the available therapeutic options [8].

The antimicrobial resistance profile of uropathogens can hinder the optimal treatment of hospitalized patients. As a result, knowing the local antimicrobial resistance rates is of great importance when recommending empirical treatment [26,27]. This highlights the need to carry out surveillance of the bacterial spectrum and AMR rates during empirical treatment of patients [27]. 

According to our findings, *E. coli* was the most common pathogen (5430 isolates), followed by *K. pneumoniae* (2302) and *E. faecalis* (1537). The overall antimicrobial resistance, after showing a small increase during the first semester of 2019, presented a continuous decrease until the end of 2020 (Figure 8). This could be explained partially by the outbreak of the COVID-19 pandemic in 2020, which led to a global lockdown and, consequently, to significantly less patients hospitalized, in general, worldwide [28]. On the contrary, a study by Mareș et al. [29], conducted in Romania, showed an increased resistance of pathogens to various antimicrobials during the COVID pandemic.

The resistance levels against extended-spectrum cephalosporins are an indicator for the production of extended spectrum beta-lactamases (ESBL), which render the treatment of *Escherichia coli* and *Klebsiella pneumoniae* infections extremely challenging [30]. In a Hungarian study by Gajdács et al. [31], the presence of ESBL isolates was reported in 8.85–38.97% and 10.89–36.06% of inpatient *E. coli* and *Klebsiella* samples, respectively. In addition, they reported a 15–40% resistance rate of *K. pneumoniae* to third-generation cephalosporins. Another study that was conducted in Romania by Chibelean et al. also presented a near 40% resistance rate of *Klebsiella* spp. to third-generation cephalosporins [32]. Our data show the average resistance rate of *E. coli* and *K. pneumoniae* to be lower than the global average rate [33], reporting, however, a *K. pneumoniae* resistance rate of 39% to ceftriaxone and over 62% to cefepime (fourth-generation cephalosporin), while the *E. coli* strains showcased under 20% resistance rates to third-generation cephalosporins (15.72% to ceftriaxone and 17.85% to ceftazidime). In a study by Ong et al. from the UK, the overall antimicrobial resistance of *E. coli* strains remained stable over the course of six years. However, there was an increase in the piperacillin/tazobactam resistance rate of 16.56%, while an increasing trend was also noticed in the resistance rate of *E. coli* to ciprofloxacin [34].

In addition, the overuse of antibiotics in the community, such as fluoroquinolones, contributes greatly to the emergence of ESBL pathogens, as well as the emergence of fluoroquinolone-resistant strains, such as *P. aeruginosa*, while keeping in mind that a great deal of fluoroquinolone overuse is still due to its use as a prophylaxis in urological procedures [35]. In a study by Benkő et al. that took place in Hungary, the ciprofloxacin resistance rate of *P. aeruginosa* isolates exceeded 20% [36]. Our study presents nearly 40% ciprofloxacin-resistant *P. aeruginosa* strains over the two-year period, while the overall resistance rate of ciprofloxacin ranged from 61.86% in the first semester of 2019 to 27.8% in the second semester of 2020 (*p*-value 0.035).

Regarding the *Enterococcal* strains, they have caused increasing concern in recent years due to their potency to spread in hospital settings among healthcare workers and patients and have been identified among the main pathogens that cause nosocomial infections. In our study, we identified a higher prevalence of *E. faecalis* than *E. faecium* in accordance with other studies [37]. Our data showed high sensitivity rates to both linezolid and vancomycin, thus identifying *Vancomycin-resistant Enterococci* (VRE) at approximately 20% for *E. faecium* and under 1% for *E. faecalis*. In recent years, *E. faecium* has substantially emerged as a source of hospital-associated infections. This is due to a variety of natural antibiotic resistance mechanisms displayed by this pathogen. Furthermore, this species is capable of acquiring resistance by mutations or by the inclusion of genes contained on mobile genetic elements such as plasmids, transposons, or integrons [38]. Kot et al. [39] showcase data from Poland, presenting multi-drug resistance in all *E. faecium* isolates, including resistance to ampicillin, imipenem, and ciprofloxacin. In addition, 40% of the isolates were resistant to glycopeptides (VRE) and aminoglycosides (gentamycin). Other European surveillance data report a frequency rate of 6.1% for VRE strains, and an alarming increase from 11.6% in 2016 to 16.8% in 2020 in the mean percentage of *vancomycin-resistant E. faecium* strains [7,40].

Urinary tract infections (UTI) are also frequently associated with urinary calculi, both as a consequence of a metabolic stone and as a cause of an infection stone due to the creation of struvite crystals by urease-producing bacterial strains. [41]. They affect almost one-third of individuals with stone disease, with the most frequent pathogenic species being *Escherichia coli*, *Klebsiella* species, *Pseudomonas aeruginosa*, *Proteus* species, and Gram-positive *cocci* in up to 25% of all cases [42]. These infections can very frequently cause fever, systemic inflammatory reaction, and sepsis. As a result, the selection of appropriate antimicrobial agents in urolithiasis patients is difficult, taking into account their polymicrobial nature, the widespread antibiotic use, and the emerging antimicrobial resistance patterns [43]. A study by De lorenzis et al. concerning urolithiasis patients showed an estimated resistance rate to fluoroquinolones of 39% for *E. coli* isolates, and over 50% for *Pseudomonas* isolates. Furthermore, it showed over 50% resistance rates of *Klebsiella* spp. isolates to cephalosporins, while 21% of *E. coli* isolates were identified as ESBL [44]. 

The advent of enormous volumes of medical data, combined with high skills in applied computer science, has resulted in the use of artificial intelligence (AI) approaches in various fields of medical research. AI has been used to diagnose calculi, forecast their composition, and predict surgical results in stone disease. AI has gained traction in microbiology, aided by advances in the genomic analysis of all types of microorganisms, and is now being researched for identifying drug and vaccine candidates, diagnosing pathogenic microorganisms, and predicting antimicrobial resistance and interactions [45,46]. The current microbiology lab capabilities require about 2 h in order for Gram-stain identification, over 24 h for microorganism identification, and up to 2 days in order to deliver an antibiotic sensitivity report. This process is time-consuming, and time is of the upmost importance, as in the case of a septic patient, for example, where an early knowledge of the susceptibilities of pathogens to antibiotics can be lifesaving [46]. A study conducted in Greece performed machine learning techniques in order to predict the antimicrobial profile of patients with stone disease with positive urine or blood cultures. It showed the feasibility of using a model to predict sensitivity to antibiotics early, when either the Gram-stain or the pathogen is known [47]. 

As for the limitations of this present study, we have to state that our data are based on the time period of 2019–2020 and not the more recent period of 2021–2022, as data are made available every two years according to the practice of the microbiology department of our hospital. We chose to include this dataset in our study as we opted to investigate the AMR profile of our department in a pre-COVID-19 era.

## 5. Conclusions

The antimicrobial resistance profiles of the most encountered pathogens in our department generally concur with relevant global-level data. The emergence of resistant pathogens has negative impacts, causing increased morbidity in patients with limited therapeutic options and economic consequences by heavily burdening all healthcare systems. This states the ever-increasing need for implementing surveillance programs in order to identify the susceptibility patterns of pathogens and to develop preventive strategies to optimize the outcomes of patients.

## Figures and Tables

**Figure 1 jcm-12-03180-f001:**
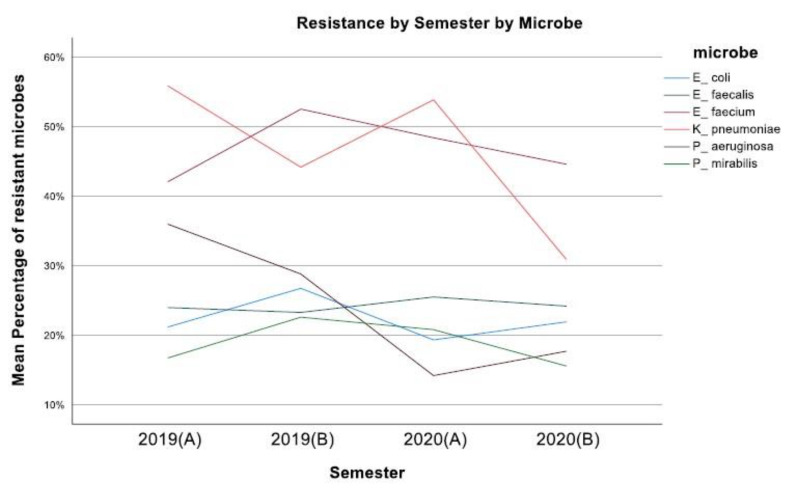
Mean resistance of microbes during the four semesters (2019–2020).

**Figure 2 jcm-12-03180-f002:**
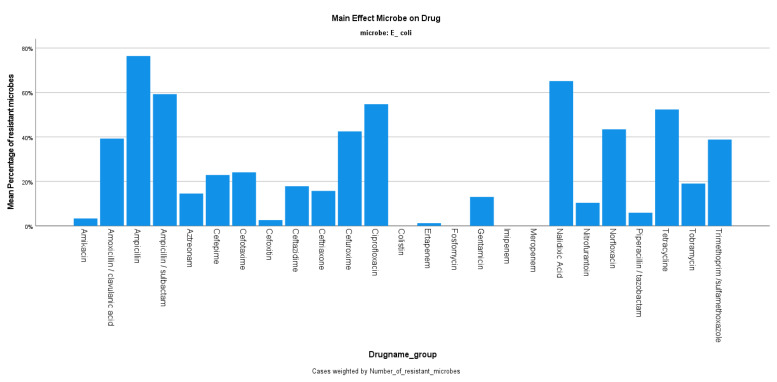
Resistance of *E. coli* against tested antibiotics.

**Figure 3 jcm-12-03180-f003:**
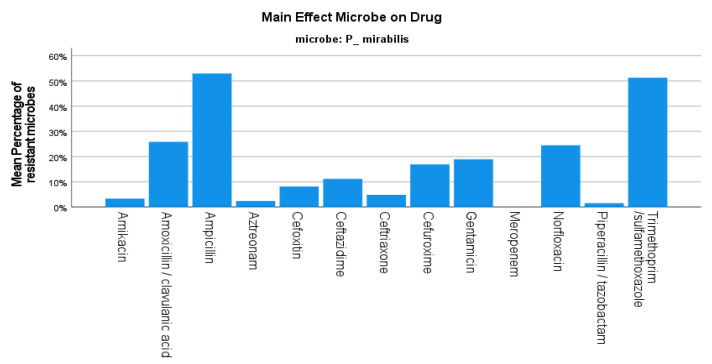
Resistance of *P. mirabilis* against tested antibiotics.

**Figure 4 jcm-12-03180-f004:**
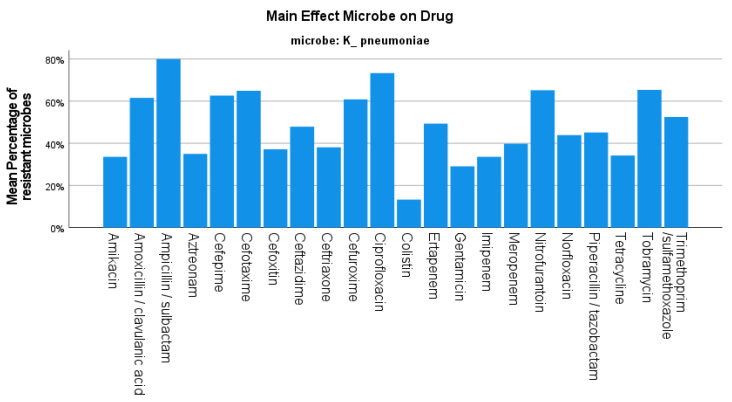
Resistance of *K. pneumoniae* against tested antibiotics.

**Figure 5 jcm-12-03180-f005:**
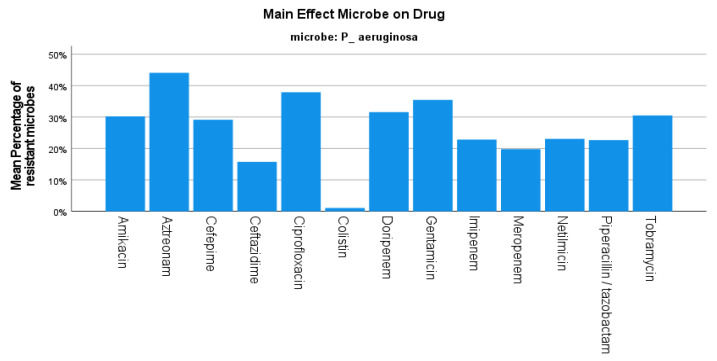
Resistance of *P. aeruginosa* against tested antibiotics.

**Figure 6 jcm-12-03180-f006:**
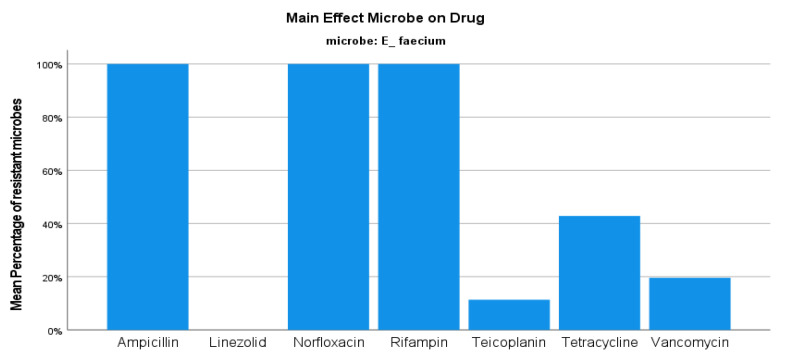
Resistance of *E. faecium* against tested antibiotics.

**Figure 7 jcm-12-03180-f007:**
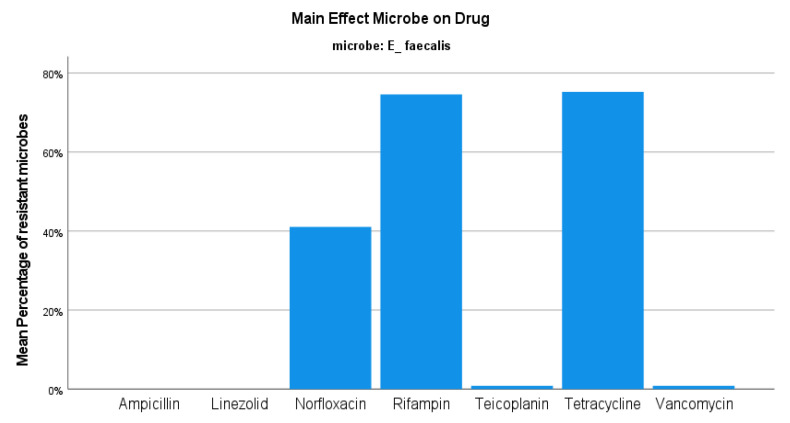
Resistance of *E. faecalis* against tested antibiotics.

**Figure 8 jcm-12-03180-f008:**
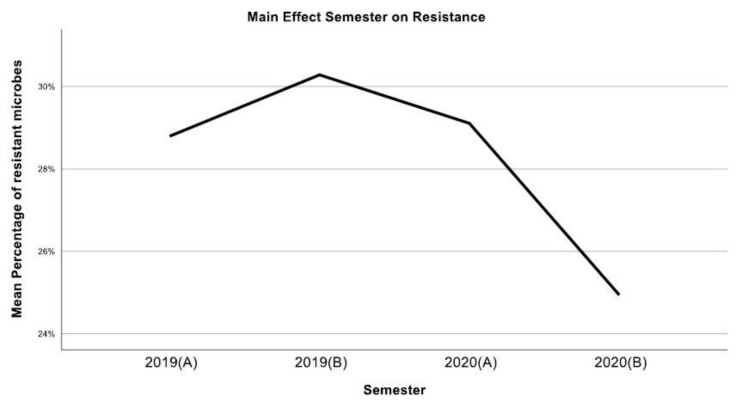
Overall resistance rates over the four semesters.

## Data Availability

Data is unavailable due to ethical restrictions.

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
