# Peer review of "A 2-Year Audit on Antibiotic Resistance Patterns from a Urology Department in Greece"

_jcm, 2023, doi:10.3390/jcm12093180_

Round 1

Reviewer 1 Report

Dear authors,

After the review process, I have some comments. Please go through the PDF manuscript and revised all the comments and suggestions briefly.

Author Response

‘’After the review process, I have some comments. Please go through the PDF manuscript and revised all the comments and suggestions briefly.’’

We thank the Reviewer for his effort in reviewing our manuscript. We went through the PDF manuscript with the highlighted sections requiring revisions point by point, and revised them.

            -  Abstract: ‘’ Put scientific names in italic. For the first appearance, write without abbreviation. ‘’

             We addressed this issue according to the reviewer’s comment.

    -     Methods: ‘’ Be more specific giving the parameters considers for the selection.’’

We addressed the specific comment on the methods section of the abstract by stating: ‘’…. as these are the most commonly encountered microbes in a Urology department.’’

  • P- value 0.000: We corrected this statement across the whole manuscript.
  • Introduction: ‘’ The problem seems not well presented. Improve it.’’

We updated the introduction section with the following: ‘’…. Some of the reasons why Greece ranks high in antibiotic-resistant bacteria, are self-medication, as antibiotics are acquired very often from pharmacies, without any prescription. In addition, over-prescription of antibiotics by Greek physicians is a major issue that leads to antibiotic over-consumption and consequently, to an increasing antimicrobial resistance [6,7].’’, in order to present the issue in its current status, in a clearer way.

  • Study design: ‘’ Provide more informations on the used antibiotics.’’

We updated our manuscript in the study design section, according to the reviewer’s comment, by presenting the antibiotics used in a classification order, to allow for a better comprehension, for instance: ’’… was measured in P. aeruginosa: aminoglycosides (amikacin, gentamycin, tobramycin), fluoroquinolones (ciprofloxacin, levofloxacin), cephalosporins (cefepime, ceftazidime), monobactams (aztreonam), carbapenems (doripenem, meropenem, imipenem), piperacillin/tazobactam (anti-pseudomonal penicillin)…’’

  • Source of isolates: ‘’ which methods?’’

    We updated the specific section of our manuscript with the following: ‘’… Isolation and identification of pathogens was carried out according to classical microbiological procedures, such as conventional cultivation-based enrichment and plating methods with rapid plating solution [14].’’

  • ‘’ there is a need of more information for this part.’’

Antimicrobial Susceptibility testing: ‘’…PCR for blaNDM, blaVIM, blaKPC, and blaOXA-48 genes.’’

We revised the specific section of our manuscript with the following; “… PCR testing was performed for the detection of the most clinically significant carbapenemase genes (blaNDM, blaVIM, blaKPC, blaOXA-48), as they encode enzymes that may generate carbapenem and other beta-lactam resistance…’’

  • Figure 1. Mean resistance of microbes during the four semesters (2019-2020).

‘’The way this result is presented is not good.  It difficult to understand those results.’’    We apologise to the reviewer for this issue. We tried to present the resistance rates of the specific pathogens in a linear fashion across the for semesters, and we are confident that by reconfiguring the figures 2-7, we will render figure 1 more understandable for the reader.

  • Discussion:

‘’ improve this section by giving more explanations after comparison.’’

  We revised the discussion section according to the reviewer’s comment as following: ‘’…. According to a study conducted by the European Centre for Disease Prevention and Control (ECDC), more than 50% of E. coli, and over 30% of K. pneumoniae isolates were resistant to at least one antimicrobial group under monitoring, … In a study by Gajdács et al [27], the presence of ESBL isolates was reported in 8.85- 38.97% and 10.89-36.06% of inpatient E. coli and Klebsiella samples, respectively. Our data show the average resistance rate of E. coli and K. pneumoniae to be lower than the global average rate [28], … In a study by BenkÅ‘ et al, ciprofloxacin resistance rate of P. aeruginosa isolates exceeded 20% [31]. Our study manifests a near 40% ciprofloxacin-resistant P. aeruginosa strains… thus identifying Vancomycin-resistant Enterococci (VRE) to approximately 20% for E. faecium and under 1% for E. faecalis, while European surveillance data report a frequency rate of 6.1 % for VRE strains, and an alarming increase from 11.6% in 2016 to 16.8% in 2020, in the mean percentage of vancomycin-resistant E. faecium strains…”.

              ‘’ there is not a need to introduce and give the aim of your work here’’

 We removed the specific highlighted part of the manuscript according to the reviewer’s comment.

  • ‘’ Fig 9 is not clear. In addition, figures should be given in the results part’’

We updated the manuscript and removed figure 9, as it was not clearly presented and we decided that the remaining figures can deliver our results in a clear way, to the readers.

‘’Follow the guidelines of the journal.’’ We revised the references section according to the journal’s guidelines and we thank the reviewer for noticing our mistake.

Reviewer 2 Report

Abstract:

1) "The high incidence of urinary tract infections (UTI), often in nosocomial environments, is a major cause for antimicrobial resistance (AMR) and the dissemination of antibiotic-resistant infections with a high health and economic burden for patients and healthcare systems, respectively": Please break this up into two sentences, since the run-on structure is hard to follow. Please make it a point to read through the rest of the manuscript and handle other run-on sentences similarly.

2) The study should mention that it is taking place in Greece.

3) "During the period 2019-2020, we included 12,215 clinical samples of blood and urine speciments, tested positive for the following pathogens: E. coli, E. faecium, E. faecalis, P. mirabilis, K. pneumoniae or P. aeruginosa.": Firstly, this sentence does not make grammatical sense and the spelling of specimens is incorrect.

4) "p-value 0.000": Is p-value = or < 0.000. I see this issue in the main-text a well. Please correct it.

Introduction

5) I see many run-on sentences in the Introduction. Please fix.

6) Can you list some reasons for Greece having a high rate of antimicrobial resistance.

Methodology:

7) "proceeded" should be processed

8) Perhaps all the information written in "2.3. Antimicrobial Susceptibility testing" may be better summarized in a Table.

Results

9) "All analyses were performed using SPSS Statistics version 24.0 (IBM, Armonk, NY, USA) [19]. The data were analyzed over a period of two years (2019-2020) and each semester was analyzed in a separate manner. ": This is more appropriate in the Methods under Statistical Analysis.

10) Please remake figures 2-7 as bar charts, as the line charts should only be used to demonstrate a temporal trend.

Discussion:

11) Please add more data from other countries in different settings, so that your results may be more contextually intepretable.

Author Response

Abstract:

  • "The high incidence of urinary tract infections (UTI), often in nosocomial environments, is a major cause for antimicrobial resistance (AMR) and the dissemination of antibiotic-resistant infections with a high health and economic burden for patients and healthcare systems, respectively": Please break this up into two sentences, since the run-on structure is hard to follow. Please make it a point to read through the rest of the manuscript and handle other run-on sentences similarly.’’

We revised the whole manuscript according the reviewer’s comment and handled all run-on sentences.

  • ‘’ The study should mention that it is taking place in Greece.’’

We mention in our study that the current study takes place in a General Hospital in Athens, Greece.

3) "During the period 2019-2020, we included 12,215 clinical samples of blood and urine speciments, tested positive for the following pathogens: E. coli, E. faecium, E. faecalis, P. mirabilis, K. pneumoniae or P. aeruginosa.": Firstly, this sentence does not make grammatical sense and the spelling of specimens is incorrect.

We thank the reviewer for pointing out our mistake. We updated our manuscript accordingly

4) ‘’"p-value 0.000": Is p-value = or < 0.000. I see this issue in the main-text a well. Please correct it.’’

We corrected our text, regarding this issue and we thank the reviewer for this comment.

Introduction

5) ‘’I see many run-on sentences in the Introduction. Please fix.’’

We updated our introduction section according to our reviewer’s comment in order to correct the issue of run-on sentences.

6) ‘’Can you list some reasons for Greece having a high rate of antimicrobial resistance.’’

We thank our reviewer for this important comment. We have updated our introduction section with the following: ‘’… Some of the reasons why Greece ranks high in antibiotic-resistant bacteria, are self-medication, as antibiotics are acquired very often from pharmacies, without any prescription. In addition, over-prescription of antibiotics by Greek physicians is a major issue that leads to antibiotic over-consumption and consequently, to an increasing antimicrobial resistance [6,7].’’

Methodology:

7) "proceeded" should be processed

We have corrected the above mistake in our manuscript.

8) ‘’Perhaps all the information written in "2.3. Antimicrobial Susceptibility testing" may be better summarized in a Table.’’

We thank our reviewer for this comment. We updated the specific section in a shorter and more comprehensive order, choosing however to keep it as text, as we believe that this way, the information it contains are more thoroughly presented.

Results:

9) "All analyses were performed using SPSS Statistics version 24.0 (IBM, Armonk, NY, USA) [19]. The data were analyzed over a period of two years (2019-2020) and each semester was analyzed in a separate manner. ": This is more appropriate in the Methods under Statistical Analysis.’’

We updated the manuscript and placed the specific section in the Methods section

10) ‘’Please remake figures 2-7 as bar charts, as the line charts should only be used to demonstrate a temporal trend.’’

We thank our reviewer for this comment, and we have replaced figures 2-7 as requested.

Discussion:

11) ‘’Please add more data from other countries in different settings, so that your results may be more contextually intepretable.’’

We thank the reviewer for the comment and we have updated the Discussion section with AMR data from other countries: “…In a study by Ong et al, the overall antimicrobial resistance of E. coli strains remained stable over the course of six years. However, there was an increase in piperacillin/tazobactam resistance rate by 16.56%, while an increasing trend was also noticed, in the resistance rate of E. coli to ciprofloxacin [29].”

Reviewer 3 Report

Dear Authors,

The manuscript submitted by Manolitsis I et al. is interesting. Antibiotic resistance leads to higher medical costs, prolonged hospital stays, and increased mortalitySurveillance is essential to inform policies and infection prevention and control responses. There are no recent reports about the AMR of uropathogens in Greek populations.

The manuscript needs significant aspects to be addressed before it can be processed further.

1. l. 19 Please write the species' name in italics E. coli, E. Faecium...Please correct the whole manuscript

2. Materials and Methods - The inclusion and exclusion criteria, such as the positive urine culture, are not defined.

3. l. 57 Please specify the name of the hospital and the city

4. l. 85 the authors should define when they include blood specimens. All the patients were hospitalized? The age of the patients and their sex should be specified.

5. l. 124 please define the mean resistance

6. l. 182-189 The authors should include data from the recent report Antimicrobial resistance in the EU/EEA (EARS-Net) - Annual Epidemiological Report for 2020

https://www.ecdc.europa.eu/en/publications-data/antimicrobial-resistance-eueea-ears-net-annual-epidemiological-report-2020

7. The discussion sections need to be revised. The authors should compare the results obtained regarding uropathogens and AMR with other recent studies from Central and Eastern Europe. (for example: 

    • PMCID: PMC6681214  

DOI: 10.3390/medicina55070356

    • PMCID: PMC7175163
    •  
    • DOI: 10.3390/life10020016

    • PMCID: PMC7560131
    •  
    • DOI: 10.3390/antibiotics9090624

    • PMCID: PMC7357063
    •  
    • DOI: 10.3390/microorganisms8060848

    • PMCID: PMC7459805
    •  
    • DOI: 10.3390/antibiotics9080472

    • PMCID: PMC8944623
    •  
    • DOI: 10.3390/antibiotics11030376

    8. Most references included in the study are at least five years old. Please update to more recent studies

    Author Response

    ‘’The manuscript needs significant aspects to be addressed before it can be processed further.’’

    1. ‘’l. 19 Please write the species' name in italics  coliE. Faecium...Please correct the whole manuscript.’’

    We have corrected the manuscript according to our reviewer’s comment.

    1. ‘’Materials and Methods - The inclusion and exclusion criteria, such as the positive urine culture, are not defined.’’

    We thank our reviewer for the comment. We updated the specific section as follows:’’… The inclusion criteria of this study included clinical isolates from non-identical, hospitalized patients that were not duplicated. On the other hand, we excluded all isolates that originated from a single patient with similar susceptibility patterns.’’

    1. ‘’l. 57 Please specify the name of the hospital and the city.’’

    We specify these data in our manuscript as requested by our reviewer.

    1. ‘’l. 85 the authors should define when they include blood specimens. All the patients were hospitalized? The age of the patients and their sex should be specified.’’

    We thank our reviewer for this comment. In our manuscript we state that all patients were hospitalized and we have updated the results section with the following: ‘’… The mean age of our patients was 67.7 years of age, and regarding their sex, 59.2% of them were males and the rest 40.8% was female patients.’’

    5.’’ l. 124 please define the mean resistance’’

    We thank our reviewer for this important comment. We updated our Materials and Methods section accordingly: ‘’… The measure of mean resistance of any microbe, is defined as the percentage of samples that are resistant to this specific microbe, to any of the antibiotics that were tested in the laboratory.’’

    6: ‘’l. 182-189 The authors should include data from the recent report Antimicrobial resistance in the EU/EEA (EARS-Net) - Annual Epidemiological Report for 2020 ’’

    https://www.ecdc.europa.eu/en/publications-data/antimicrobial-resistance-eueea-ears-net-annual-epidemiological-report-2020

    As requested, we have updated our text with data from the report above, in the Discussion section : ‘’… According to a study conducted by the European Centre for Disease Prevention and Control (ECDC), more than 50% of E. coli, and over 30% of K. pneumoniae isolates were resistant to at least one antimicrobial group under monitoring, and combination resistance to multiple antimicrobial groups was common. Consequently, approximately 33,000 people die each year as a result of an infection caused by bacteria resistant to antibiotics… and an alarming increase from 11.6% in 2016 to 16.8% in 2020, in the mean percentage of vancomycin-resistant E. faecium strains’’

    1. ‘’The discussion sections need to be revised. The authors should compare the results obtained regarding uropathogens and AMR with other recent studies from Central and Eastern Europe. ‘’

    We thank our reviewer for this comment. We have updated our Discussion section with data from recent studies, pointed by our reviewer as examples: ‘’…. In a study by Gajdács et al [27], the presence of ESBL isolates was reported in 8.85- 38.97% and 10.89-36.06% of inpatient E. coli and Klebsiella samples, respectively…In a study by BenkÅ‘ et al, ciprofloxacin resistance rate of P. aeruginosa isolates exceeded 20%...’’

    1. ‘’Most references included in the study are at least five years old. Please update to more recent studies.’’

    We thank our reviewer for this very important comment. For that reason, we have updated our manuscript with more recent studies as references. Please see references no. 4,27,29,31.

    Round 2

    Reviewer 1 Report

    well written.

    Author Response

    Dear Editor and Reviewers,

    We wish to express our sincere gratitude for your willingness and effort. All comments helped towards improving the scientific integrity and readability of our manuscript.

    Please find below our response to the comments point-by-point with references to the revised text for your ease.

    We remain at your disposal for any further action needed from our side.

    With kind regards,

    Tzelves Lazaros

    Second Department of Urology, Sismanogleio General Hospital, Marousi, Greece

    Reviewer 3 (round 2)

    • “2. Materials and Methods - Study design - it is unclear what the authors understand by clinical isolates; please define positive urine culture. The authors still have not specified the criteria for blood specimens. Did the authors include patients with urinary catheters (cUtis)?”

    We thank our reviewer for the comment. We have updated the Source of Isolates section appropriately:” … Positive urine culture…department.” In our study we included all patients, regardless of having a urinary catheter or not, as the identification of urinary pathogens is important for administering appropriate antibiotic prophylaxis during operations that breach the urothelial mucosa (transurethral prostatectomy for example)

    • “l. 117 references 10 it is not appropriate”

    We have removed the specific reference. 

    • “Fig 2 is not clear, there are blue bars without antibiotics”

    We have corrected the specific figure.

    • Figure 8 - Overall resistance of who?

    Figure 8 presents the trend of overall resistance rates of the selected pathogens against tested antibiotics, in our department, over the course of four semesters.

    • ‘’Some comments from my previous review have not been addressed properly:

    "The discussion sections need to be revised. The authors should compare the results obtained regarding uropathogens and AMR with other recent studies from Central and Eastern Europe. "

    The discussion section needs significant revision. The authors have included only four sentences; they must include the country for other studies mentioned. They have to integrate and compare their results with other European countries, not just include one study for one species. 

    For the discussion section, each paragraph should have an opening statement defining the topic of the discussion e.g. resistance to E. coli, then citing relevant researches supporting/contradicting their findings, identifying or correlating limitations of their present study, and conclusions or take-home message from the discussion. Most conclusions drawn are speculative and not supported by the data.

    "Most references included in the study are at least five years old. Please update to more recent studies." The authors should try to cite more recent data.’’

      We thank our reviewer very much for the important comments. We thus, have updated our manuscript accordingly, compared our results with other studies from central Europe, included the country of origin of the studies mentioned, and tried to update our paper with as more recent studies as possible (please see references no. 4,13,26,28,29,33). We refer to the following from our manuscript:” … On the contrary, a study by MareÈ™ et al [26], conducted in Romania, … the presence of ESBL isolates in this study from Hungary… Another study that was conducted in Romania… In a study by Ong et al from the UK…”. We believe that our conclusions concur with our results and are in accordance to international data.

    Reviewer 3 Report

    Dear Authors,

    First of all, I want to congratulate you on your work. The manuscript is improved. Some aspects are still unclear.

    2. Materials and Methods - Study design - it is unclear what the authors understand by clinical isolates; please define positive urine culture. The authors still have not specified the criteria for blood specimens. Did the authors include patients with urinary catheters (cUtis)?

    l. 117 references 10 it is not appropriate 

    Fig 2 is not clear, there are blue bars without antibiotics

    Figure 8 - Overall resistance of who?

    Some comments from my previous review have not been addressed properly:

    "The discussion sections need to be revised. The authors should compare the results obtained regarding uropathogens and AMR with other recent studies from Central and Eastern Europe. "

    The discussion section needs significant revision. The authors have included only four sentences; they must include the country for other studies mentioned. They have to integrate and compare their results with other European countries, not just include one study for one species. 

    For the discussion section, each paragraph should have an opening statement defining the topic of the discussion e.g. resistance to E. coli, then citing relevant researches supporting/contradicting their findings, identifying or correlating limitations of their present study, and conclusions or take-home message from the discussion.

    Most conclusions drawn are speculative and not supported by the data.

    "Most references included in the study are at least five years old. Please update to more recent studies." The authors should try to cite more recent data.

    Author Response

    Dear Editor and Reviewers,

    We wish to express our sincere gratitude for your willingness and effort in considering for publication our manuscript at your prestigious journal. All comments helped towards improving the scientific integrity and readability of our manuscript.

    Please find below our response to the comments point-by-point with references to the revised text for your ease.

    We remain at your disposal for any further action needed from our side.

    With kind regards,

    Tzelves Lazaros

    Second Department of Urology, Sismanogleio General Hospital, Marousi, Greece

    Reviewer 3 (round 2)

    • “2. Materials and Methods - Study design - it is unclear what the authors understand by clinical isolates; please define positive urine culture. The authors still have not specified the criteria for blood specimens. Did the authors include patients with urinary catheters (cUtis)?”

    We thank our reviewer for the comment. We have updated the Source of Isolates section appropriately:” … Positive urine culture…department.” In our study we included all patients, regardless of having a urinary catheter or not, as the identification of urinary pathogens is important for administering appropriate antibiotic prophylaxis during operations that breach the urothelial mucosa (transurethral prostatectomy for example)

    • “l. 117 references 10 it is not appropriate”

    We have removed the specific reference. 

    • “Fig 2 is not clear, there are blue bars without antibiotics”

    We have corrected the specific figure.

    • Figure 8 - Overall resistance of who?

    Figure 8 presents the trend of overall resistance rates of the selected pathogens against tested antibiotics, in our department, over the course of four semesters.

    • ‘’Some comments from my previous review have not been addressed properly:

    "The discussion sections need to be revised. The authors should compare the results obtained regarding uropathogens and AMR with other recent studies from Central and Eastern Europe. "

    The discussion section needs significant revision. The authors have included only four sentences; they must include the country for other studies mentioned. They have to integrate and compare their results with other European countries, not just include one study for one species. 

    For the discussion section, each paragraph should have an opening statement defining the topic of the discussion e.g. resistance to E. coli, then citing relevant researches supporting/contradicting their findings, identifying or correlating limitations of their present study, and conclusions or take-home message from the discussion. Most conclusions drawn are speculative and not supported by the data.

    "Most references included in the study are at least five years old. Please update to more recent studies." The authors should try to cite more recent data.’’

      We thank our reviewer very much for the important comments. We thus, have updated our manuscript accordingly, compared our results with other studies from central Europe, included the country of origin of the studies mentioned, and tried to update our paper with as more recent studies as possible (please see references no. 4,13,26,28,29,33). We refer to the following from our manuscript:” … On the contrary, a study by MareÈ™ et al [26], conducted in Romania, … the presence of ESBL isolates in this study from Hungary… Another study that was conducted in Romania… In a study by Ong et al from the UK…”. We believe that our conclusions concur with our results and are in accordance to international data.